# Screening for Latent Tuberculosis Across Chronic Kidney Disease Stages Using Interferon-Gamma Release Assay: Findings from a National Infectious Disease Institute in Thailand

**DOI:** 10.3390/tropicalmed10080235

**Published:** 2025-08-20

**Authors:** Wannarat Pongpirul, Krit Pongpirul, Vongsatorn Tiabrat, Karnsuwee Muennoo, Wisit Prasithsirikul

**Affiliations:** 1Bamrasnaradura Infectious Diseases Institute, Department of Disease Control, Ministry of Public Health, Nonthaburi 11000, Thailand; 2Center of Excellence in Preventive and Integrative Medicine, Faculty of Medicine, Chulalongkorn University, Bangkok 10330, Thailand; 3Department of Infection Biology & Microbiomes, Faculty of Health and Life Sciences, University of Liverpool, Liverpool L69 3GE, UK; 4Clinical Research Center, Bumrungrad International Hospital, Bangkok 10110, Thailand

**Keywords:** chronic kidney disease, CKD stages, eGFR, indeterminate result, interferon-gamma release assay, latent tuberculosis infection, QuantiFERON-TB Gold, Thailand

## Abstract

Background: Latent tuberculosis infection (LTBI) is a major global health concern, particularly among individuals with chronic kidney disease (CKD), who are at increased risk of reactivation due to impaired immunity and frequent exposure to immunosuppressive therapies. Despite growing reliance on interferon-gamma release assays (IGRAs) such as QuantiFERON-TB Gold In-Tube (QFT-GIT) in BCG-vaccinated populations, data on IGRA performance across CKD stages remain limited in resource-limited settings. Objective: To determine the prevalence of LTBI and indeterminate IGRA results across CKD stages in a Thai population and assess the clinical utility of IGRA in this context. Materials and Methods: We conducted a cross-sectional study among 785 Thai adults receiving care at a national infectious disease institute, including diabetes clinic patients, hospital staff, and individuals on hemodialysis. Each participant underwent QFT-GIT testing, and the CKD stage was classified using the estimated glomerular filtration rate (eGFR) closest prior to testing. Results: Overall IGRA positivity was 22.2%, peaking in CKD stage G3 (31.6%) and declining in stage G5 (11.0%), where indeterminate results were also highest (6.8%). Limitations: Single-center design and lack of confirmatory testing may limit generalizability. Conclusions: IGRA performance is reliable in early-to-moderate CKD but less so in advanced stages. LTBI is prevalent in CKD stages G2–G4, supporting stage-specific approaches to LTBI screening and caution against overreliance on IGRA in advanced renal impairment.

## 1. Introduction

Tuberculosis (TB) remains a major global health threat, with an estimated 10.6 million new cases and 1.3 million deaths in 2022 [1]. Latent tuberculosis infection (LTBI) is defined by the World Health Organization (WHO) as a state of persistent immune response to *Mycobacterium tuberculosis* (MTB) antigens without evidence of active disease [2]. Globally, approximately one-quarter of the population is infected, serving as a critical reservoir for future TB cases and forming a central focus of global TB elimination strategies [2,3]. The risk of progression from LTBI to active TB is 5–10% over a lifetime but is markedly higher in high-risk groups such as children, people living with HIV, and those with chronic kidney disease (CKD) [2].

Individuals with CKD are especially vulnerable to TB reactivation due to a combination of uremia-induced immune dysfunction, systemic inflammation, impaired T-cell-mediated immunity, and exposure to immunosuppressive medications [4,5,6,7]. A recent meta-analysis by Alemu et al. (2023) estimated a pooled LTBI prevalence of 34.8% among patients with CKD, with variation based on diagnostic modality, CKD stage, and geographic region [8]. In parallel, Zhou et al. (2023) demonstrated that individuals with diabetes mellitus (DM)—a common comorbidity in CKD—had an overall LTBI prevalence of 33.2%, reinforcing the intersection between TB, DM, and CKD [9]. These findings underscore the importance of integrated TB screening strategies in non-communicable disease (NCD) populations.

Traditionally, LTBI screening has relied on the tuberculin skin test (TST), which has poor specificity in BCG-vaccinated populations and reduced sensitivity in immunocompromised hosts [10,11,12]. Interferon-gamma release assays (IGRAs), including QuantiFERON-TB Gold In-Tube (QFT-GIT), QuantiFERON-TB Gold Plus (QFT-Plus), and T-SPOT.TB, offer improved specificity by targeting TB-specific antigens not shared with BCG or most environmental mycobacteria [13]. A systematic review by Shafeque et al. (2020) reported a pooled sensitivity and specificity for QFT-GIT of 76% and 98%, respectively, in global populations [14]. However, test performance may degrade in advanced CKD due to lymphopenia and impaired cytokine responses, resulting in higher indeterminate rates [4,15,16].

Connell et al. (2011) found that QFT-GIT and T-SPOT.TB showed higher concordance and sensitivity than TST in CKD patients, with T-SPOT.TB demonstrating fewer indeterminate results [16]. Dyrhol-Riise et al. (2010) similarly observed a better IGRA performance than TST in immunosuppressed populations. While T-SPOT.TB may be more resilient in lymphopenic states, its implementation in resource-limited settings may be constrained by cost and laboratory infrastructure [17]. In Thailand, BCG vaccination is universal at birth, and the TB burden remains high. Klayut et al. (2024) reported a 20.6% IGRA positivity rate among Thai healthcare workers using QFT-Plus, demonstrating feasibility in local settings [18]. Yet, limited data exist on IGRA performance across eGFR-defined CKD stages in Thai patients. Existing studies in Asia, such as by Shu et al. (2015) in Taiwan and Lee et al. (2015, 2021) in Korea, have suggested rising indeterminate rates with worsening kidney function and reduced IFN-γ responses in patients with eGFR < 30 mL/min/1.73 m^2^ [15,19,20].

This study aimed to fill the evidence gap regarding the performance of QFT-GIT in detecting LTBI across the spectrum of CKD stages in a Thai population. Specifically, we sought to determine the prevalence of IGRA-positive results as a marker of LTBI, quantify the frequency of indeterminate IGRA results, and explore the clinical and demographic factors associated with these outcomes. We used the estimated glomerular filtration rate (eGFR) closest prior to IGRA testing to classify CKD stages, thereby enhancing the temporal validity of kidney function assignment. This study was conducted at the Bamrasnaradura Infectious Diseases Institute (BIDI), a national reference center under the Department of Disease Control, Ministry of Public Health, Thailand, to provide insights that could inform targeted TB screening policies in CKD populations.

## 2. Materials and Methods

This was a cross-sectional, retrospective hospital-based study conducted at the Bamrasnaradura Infectious Diseases Institute (BIDI), Nonthaburi, Thailand, a national infectious disease referral hospital under the Department of Disease Control, Ministry of Public Health. The study period spanned from January to June 2025.

Eligible participants were adult aged ≥ 18 years who underwent QuantiFERON-TB Gold In-Tube (QFT-GIT) testing during the study period and had an available estimated glomerular filtration rate (eGFR) measurement within three months prior to testing. Participants were recruited from three groups: patients attending the diabetes (DM) clinic, hospital staff undergoing a health check-up, and individuals on maintenance hemodialysis (HD). Exclusion criteria included incomplete laboratory records, prior history of active tuberculosis, or indeterminate QFT-GIT results due to technical error.

Using proportional stratified random sampling, we selected 400 participants across the full spectrum of CKD stages (G1–G5) and HD, proportional to the CKD stage distribution in our hospital’s chronic disease registry. Patients with known active TB, those on current anti-TB treatment, and those with acute illness at the time of sampling were excluded.

After obtaining written informed consent, clinical and demographic data were collected from medical records and direct interviews. Variables included age, sex, BMI, comorbidities (diabetes, hypertension, dyslipidemia, cardiovascular disease, malignancy, autoimmune disease, cirrhosis, HIV status), immunosuppressive medication use, smoking history, and household TB contact history. Laboratory values including serum creatinine, estimated glomerular filtration rate (eGFR, using the CKD-EPI formula), and QuantiFERON-TB Gold In-Tube (QFT-GIT) results were extracted.

To accurately reflect kidney function at the time of IGRA testing, CKD staging was based on the eGFR value closest in time but prior to or on the same day as the IGRA test. Among up to five creatinine-based eGFR measurements available per subject, the most recent eGFR prior to the IGRA date was selected. CKD stages were defined per KDIGO 2012 guidelines as follows:G1: eGFR ≥ 90 mL/min/1.73 m^2^;G2: eGFR 60–89;G3a: eGFR 45–59;G3b: eGFR 30–44;G4: eGFR 15–29;G5: eGFR < 15 (non-dialysis);HD: currently receiving maintenance hemodialysis.

All participants underwent IGRA testing using QFT-GIT (Qiagen, Hilden, Germany), performed at the National TB Reference Laboratory. Interpretation followed manufacturer’s guidelines:Positive: IFN-γ TB antigen minus nil ≥ 0.35 IU/mL and ≥25% of nil;Negative: IFN-γ TB antigen minus nil < 0.35 IU/mL or <25% of nil, with adequate mitogen response;Indeterminate: Inadequate mitogen response (<0.5 IU/mL) or elevated nil control (>8.0 IU/mL).

Descriptive statistics were used to summarize demographic and clinical characteristics overall and by CKD stage and patient category (D: DM clinic, H: HD unit, S: staff). Continuous variables were presented as mean ± standard deviation (SD) or median (IQR), as appropriate. Categorical variables were reported as frequencies and percentages. Chi-square and Fisher’s exact tests were used to compare IGRA positivity and indeterminate rates across CKD stages. Logistic regression was used to explore predictors of IGRA positivity and indeterminate results. All analyses were performed using STATA SE version 19 (StataCorp, College Station, TX, USA). A two-sided *p*-value < 0.05 was considered statistically significant.

This study was approved by the BIDI Institutional Review Board (IRB no. S001h/68).

Generative AI (ChatGPT-4, OpenAI) was used to assist in drafting and refining the text of this manuscript, specifically for summarizing the literature, editing for narrative clarity, and organizing sections such as the abstract, introduction, and discussion. It was not used to generate data, conduct analysis, design this study, or interpret findings. All content was reviewed and approved by the authors, who take full responsibility for the final manuscript.

## 3. Results

### Baseline Characteristics by Care Setting

Table 1 summarizes the demographic, clinical, and laboratory characteristics of the 785 participants, stratified by care setting: diabetes (DM) clinic (*n* = 516), hemodialysis (HD) unit (*n* = 65), and hospital staff (*n* = 204). The overall mean age was 56.9 years (SD ± 17.1), with significant variation across groups: HD patients were the oldest (mean 69.1 years), followed by DM clinic attendees (63.5 years) and hospital staff (36.2 years) (*p* < 0.0001). The proportion of female participants also differed markedly, highest among hospital staff (88.7%) and lowest in the HD unit (41.5%) (*p* < 0.0001).

Metabolic comorbidities were most prevalent in the DM and HD groups. Diabetes was nearly universal in the DM clinic (99.8%) and common in HD patients (70.8%), but rare among hospital staff (11.3%). A similar trend was observed for hypertension and dyslipidemia, both significantly more prevalent in the DM and HD groups (*p* < 0.0001 for both).

Cardiovascular disease, cerebrovascular disease, and chronic kidney disease (CKD) were significantly more common among HD patients (30.8%, 13.8%, and 92.3%, respectively) compared to other groups. Notably, none of the hospital staff had CKD. Body mass index (BMI) was significantly lower in the HD group (mean 23.4 kg/m^2^) compared to the DM clinic and staff groups (27.7 and 25.1 kg/m^2^, respectively) (*p* < 0.0001). Hemoglobin levels were also lower in the HD group (mean 10.5 g/dL), reflecting anemia common in end-stage renal disease.

Lipid profiles showed lower total cholesterol and HDL levels among HD patients, while triglyceride levels were similar across groups. Liver enzymes (AST and ALT) were significantly lower in the HD group compared to others. Inflammatory markers such as WBC and differential counts (neutrophils, lymphocytes) showed significant variation by group (*p* < 0.0001 for both neutrophils and lymphocytes), with notably lower lymphocyte percentages in the HD group.

Hepatitis B surface antibody (anti-HBs) positivity was highest among HD patients (84.6%) and hospital staff (63.2%), likely reflecting vaccination coverage, while anti-HBc and anti-HCV positivity rates remained low across all groups, with modest elevations among staff and HD patients.

These findings highlight the marked demographic and clinical heterogeneity across the three care settings and underscore the differences in comorbidity burden, immune markers, and liver and kidney function profiles.

Table 2 shows the distribution of QuantiFERON-TB Gold (IGRA) results across CKD stages, classified using the estimated glomerular filtration rate (eGFR) value closest prior to IGRA testing. Among participants with stage 1 CKD (*n* = 378), 17.2% had a positive IGRA result, with only 1.1% classified as indeterminate. IGRA positivity increased with CKD stage, peaking at 31.6% in stage 3 and 30.8% in stage 4. However, a decline was observed in stage 5, with only 11.0% positivity and a relatively high rate (6.8%) of indeterminate results. Indeterminate results were absent in stages 2 to 4, suggesting preserved immune response in these groups, while the increased rate in stage 5 may reflect immune dysfunction associated with advanced CKD. These findings suggest a non-linear relationship between kidney function and IGRA performance, with implications for latent TB screening strategies in patients with renal impairment.

## 4. Discussion

This study provides new evidence on the distribution of latent tuberculosis infection (LTBI) and the diagnostic performance of QuantiFERON-TB Gold In-Tube (QFT-GIT) across estimated glomerular filtration rate (eGFR)-based CKD stages in a tuberculosis (TB)-endemic setting. We observed a non-linear pattern of IGRA positivity, peaking in CKD stages G2 to G3b and declining in G4, accompanied by a significant increase in indeterminate results in stages G4 and G5. These findings reflect both biological limitations and technical challenges of interferon-gamma release assays (IGRAs) in immunocompromised populations and hold implications for screening strategies among high-risk groups in Thailand and similar low- and middle-income countries (LMICs).

The elevated IGRA positivity in early to moderate CKD stages (G2–G3b) aligns with previous studies, including those from Korea and Turkey, which reported positivity rates between 20% and 35% among pre-dialysis and dialysis patients [19,21]. Our study is distinctive in linking eGFR temporally to the IGRA test and stratifying patients using the most proximal eGFR result before testing, thereby minimizing stage misclassification due to fluctuating renal function. The highest LTBI prevalence in stages G2 and G3b suggests that patients with moderate CKD retain sufficient immune responsiveness to yield reliable IGRA results and may represent an optimal target group for LTBI screening and preventive therapy. This observation is consistent with WHO guidelines, which recommend targeted LTBI screening in populations with an elevated reactivation risk, including those with advanced chronic diseases [2]

However, we observed a marked increase in indeterminate IGRA results in stages G4 and G5, reflecting declining assay reliability in advanced CKD. This finding is consistent with previous reports [11,15,16], which attributed indeterminacy to impaired mitogen responses, low lymphocyte counts, hypoalbuminemia, and systemic inflammation. In our study, patients with advanced CKD also had reduced lymphocyte percentages, reinforcing the mechanistic plausibility of immunologic exhaustion affecting test validity. Current WHO guidance notes that IGRA results should be interpreted with caution in immunocompromised hosts, and our findings provide context-specific evidence supporting this recommendation [2,3]

The apparent rebound in IGRA positivity in G5 (30.8%) and HD (27.7%) may be misleading. While it could reflect a higher true LTBI burden in these groups due to repeated exposure in dialysis settings [6,7], it could also arise from non-specific background IFN-γ production, as suggested in studies documenting false-positive rates in severely immunocompromised patients [16,17]. Therefore, these results should be interpreted with caution, especially when indeterminate rates are concurrently elevated.

The strength of IGRA-based screening in BCG-vaccinated populations like Thailand lies in its improved specificity compared to the tuberculin skin test (TST) [10,14]. However, in advanced CKD, test performance degrades, and alternative approaches such as T-SPOT.TB or newer IFN-γ release platforms may offer better diagnostic reliability [17,22]. T-SPOT.TB, for example, has demonstrated lower indeterminate rates in lymphopenic patients and may be more resilient in immunosuppressed settings [5,16].

Our findings are also consistent with broader evidence linking diabetes and CKD with an elevated LTBI risk. Zhou et al. reported a pooled LTBI prevalence of 33.2% among people with diabetes, consistent with our subgroup findings [9]. Ugarte-Gil et al. further argued for integrated screening among patients with diabetes, CKD, and rheumatoid arthritis, given their shared immune vulnerabilities [23]. This aligns with calls to merge TB and non-communicable disease (NCD) programs to optimize resource use and early detection [24].

From a policy perspective, our data suggest that CKD stages G2 to G3b may be the most reliable and impactful windows for LTBI screening. In contrast, advanced CKD stages require cautious interpretation and possibly the use of confirmatory platforms or adjunctive biomarkers. This tailored approach is in line with global guidance, which advocates for context-specific adaptation of LTBI screening strategies to balance diagnostic yield with feasibility [2,3].

Strengths of this study include a large sample size with proportional representation across CKD stages and care settings, use of eGFR temporally linked to IGRA testing, and robust clinical data to adjust for confounders. The setting at a national infectious disease referral center enhances the generalizability to similar high-burden LMIC environments.

This study has several limitations. Its cross-sectional design limits the ability to establish causal relationships between CKD stage and IGRA results. We did not perform confirmatory testing with T-SPOT.TB or TST, which could have strengthened diagnostic comparisons. Potential confounding factors—such as malnutrition, undocumented immunosuppressive drug use, or latent viral co-infections—were not systematically captured. Although CKD staging was based on the eGFR measurement closest to the IGRA test, transient renal function fluctuations or intercurrent illnesses may have led to minor misclassification.

Future studies should prospectively evaluate IGRA dynamics, including test reversion or conversion and correlation with TB incidence. Comparative studies between QFT-GIT, QFT-Plus, and T-SPOT.TB in CKD populations—particularly in stages G4–G5—are urgently needed to refine diagnostic algorithms. Moreover, integrating TB screening into CKD and diabetes care workflows, supported by electronic health record flags and clinical decision tools, may enable scalable and cost-effective LTBI control in high-risk populations.

## 5. Conclusions

This study demonstrates significant variation in QuantiFERON-TB Gold In-Tube (QFT-GIT) positivity across chronic kidney disease (CKD) stages, with the highest prevalence observed in moderate renal impairment (G2–G3b, *p* < 0.05). In contrast, indeterminate result rates were significantly higher in advanced stages (G4–G5, *p* < 0.05), consistent with impaired cellular immunity and lymphopenia in these populations.

These findings support the clinical utility of IGRAs in early to moderate CKD, particularly among individuals with diabetes or other TB risk factors, while highlighting reduced reliability in advanced disease. For G4–G5 or hemodialysis patients, alternative diagnostic approaches such as T-SPOT.TB or adjunctive immune biomarkers may be warranted. Incorporating the CKD stage into national TB screening policies may improve LTBI detection and preventive treatment. Future prospective studies should compare IGRA platforms and assess the long-term outcomes of LTBI management in CKD populations.

## Figures and Tables

**Table 1 tropicalmed-10-00235-t001:** Demographic, clinical, and laboratory characteristics by care setting (DM clinic, hemodialysis unit, hospital staff).

Variable	Overall	DM Clinic	Hemodialysis Unit	Hospital Staff	*p*-Value
*n*	785	516	65	204	<0.0001
Age (years)	56.9 ± 17.1	63.5 ± 12.1	69.1 ± 14.2	36.2 ± 9.9	<0.0001
Female (%)	62.5%	54.8%	41.5%	88.7%	<0.0001
BMI (kg/m^2^)	26.7 ± 5.6	27.7 ± 5.5	23.4 ± 5.0	25.1 ± 5.5	<0.0001
Prior TB (%)	4.0%	5.2%	6.2%	0.5%	0.0082
Abnormal chest X-ray (%)	0.1%	0.0%	0.0%	0.5%	0.2403
Diabetes (%)	74.4%	99.8%	70.8%	11.3%	<0.0001
Hypertension (%)	66.1%	80.8%	93.8%	20.1%	<0.0001
Dyslipidemia (%)	62.0%	75.8%	92.3%	11.3%	<0.0001
CAD (%)	10.6%	13.0%	30.8%	0.5%	<0.0001
CVA (%)	5.9%	7.0%	13.8%	1.5%	0.0002
CKD (%)	47.6%	55.2%	92.3%	0.0%	<0.0001
Malignancy (%)	8.4%	9.7%	13.8%	2.0%	0.0002
Cirrhosis (%)	0.9%	1.2%	0.0%	0.5%	0.5089
Autoimmune (%)	1.9%	2.3%	1.5%	1.0%	0.5196
Immunosuppressive (%)	1.8%	2.1%	3.1%	1.0%	0.5111
COPD/Asthma (%)	3.4%	4.3%	4.6%	0.5%	0.0394
Thalassemia (%)	1.7%	2.3%	0.0%	1.0%	0.2424
Thyroid disorder (%)	7.1%	8.1%	4.6%	4.9%	0.3009
FBS (mg/dL)	129.0 ± 57.2	137.9 ± 59.2	134.2 ± 64.1	94.6 ± 23.5	<0.0001
HbA1c (%)	7.1 ± 1.8	7.2 ± 1.7	6.7 ± 1.8	5.7 ± 0.8	<0.0001
Hemoglobin (g/dL)	12.8 ± 1.7	13.0 ± 1.7	10.5 ± 1.4	13.1 ± 1.3	<0.0001
WBC (cells/µL)	6905.8 ± 1847.8	7067.7 ± 1880.4	6567.7 ± 2038.3	6603.9 ± 1646.6	0.0030
Neutrophil (%)	59.8 ± 9.6	59.3 ± 9.2	69.1 ± 11.1	58.1 ± 8.2	<0.0001
Lymphocyte (%)	30.3 ± 9.0	30.7 ± 8.4	18.6 ± 8.2	33.2 ± 7.7	<0.0001
Ferritin (ng/mL)	302.8 ± 263.0	248.6 ± 179.5	305.3 ± 266.9	-	-
Total cholesterol (mg/dL)	185.1 ± 43.9	164.8 ± 26.6	152.0 ± 37.9	198.8 ± 41.0	<0.0001
LDL (mg/dL)	98.3 ± 36.0	96.5 ± 33.3	95.6 ± 41.4	104.3 ± 40.6	0.0223
HDL (mg/dL)	57.9 ± 15.2	54.5 ± 10.8	47.3 ± 14.6	61.9 ± 14.3	<0.0001
Triglycerides (mg/dL)	127.6 ± 66.9	122.1 ± 45.4	117.7 ± 119.0	131.6 ± 42.2	0.3093
AST (U/L)	28.9 ± 14.5	29.6 ± 27.3	22.4 ± 10.3	30.9 ± 12.0	0.0002
ALT (U/L)	31.0 ± 17.9	27.1 ± 20.2	15.7 ± 9.3	36.5 ± 16.6	<0.0001
HBsAg-positive (%)	1.1%	1.2%	1.5%	1.0%	0.9329
Anti-HBs-positive (%)	24.1%	1.0%	84.6%	63.2%	<0.0001
Anti-HBc-positive (%)	1.7%	0.2%	1.5%	5.4%	<0.0001
Anti-HCV-positive (%)	0.5%	0.4%	3.1%	0.0%	0.0080

**Table 2 tropicalmed-10-00235-t002:** QuantiFERON-TB Gold (IGRA) results across CKD stages by eGFR closest before testing.

CKD Stage	Total (*n*)	IGRA-Positive (*n*)	IGRA-Positive (%)	IGRA-Indeterminate (*n*)	IGRA-Indeterminate (%)
Stage 1	378	65	17.2%	4	1.1%
Stage 2	216	57	26.4%	0	0.0%
Stage 3	79	25	31.6%	0	0.0%
Stage 4	39	12	30.8%	0	0.0%
Stage 5	73	8	11.0%	5	6.8%

## Data Availability

The data supporting the findings of this study are not publicly available due to ethical and institutional privacy restrictions but may be made available from the corresponding author upon reasonable request and with appropriate institutional approvals.

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
