# Peer review of "Screening for Latent Tuberculosis Across Chronic Kidney Disease Stages Using Interferon-Gamma Release Assay: Findings from a National Infectious Disease Institute in Thailand"

_tropicalmed, 2025, doi:10.3390/tropicalmed10080235_

Round 1

Reviewer 1 Report

Comments and Suggestions for Authors

Dear colleagues,

In the manuscript «Screening for Latent Tuberculosis Across CKD Stages Using IGRA: Findings from a National Infectious Diseases Institute in Thailand» colleagues have found useful and current data.

Currently, the problem of tuberculosis diagnosis remains extremely relevant. Colleagues have presented highly interesting research results in LTBI diagnosis.

The following items should be corrected:

  • - In the title of the article, the abbreviation of the term must be expanded and the period removed.
  • The structure of the abstract should be corrected and should include: background; objective; materials and methods; results; limitations, and conclusions.
  • keywords are representative of the research. They should be arranged alphabetically.
  • Epidemiologic data presented in the WHO Report 2024 should be provided in the article should be included. Global TB trends: Consider discussing whether the trend of TB incidence continued into 2023–2024 to provide up-to-date epidemiological context.
  • In the introduction relevant data about LTBI diagnosis should be applied:

Latent tuberculosis infection: updated and consolidated guidelines for programmatic management. Geneva: World Health Organization; 2018 (https://apps.who.int/iris/handle/10665/260233, accessed 23 February 2022).

- WHO consolidated guidelines on tuberculosis. Module 5: management of tuberculosis in children and adolescents. Geneva: World Health Organization; 2022.

- Jean Pierre Zellweger , Giovanni Sotgiu, Massimo Corradi, Paolo Durando. The diagnosis of latent tuberculosis infection (LTBI): currently available tests, future developments, and perspectives to eliminate tuberculosis (TB). Med Lav. 2020 Jun 26;111(3):170-183. doi: 10.23749/mdl.v111i3.9983

  • The aim of the study not clear and not presented.
  • In the chapter «Materials and methods» should be presented characteristics of the study design (prospective or retrospective study, period of time, inclusion and exclusion criteria and others).
  • Statistical methods should be provided in the chapter "Materials and Methods".
  • The specificity and sensitivity of IGRA methods in patients with comorbidities should be provided
  • The discussion chapter should be expanded. It should be written the connection between your review content and existing clinical practice guidelines (e.g., WHO, CDC). Where applicable, highlight whether your synthesis supports or challenges current recommendations
  • Limitations of the study did not present.
  • Conclusions are not clear. Colleagues did not provide statistically significant evidence of conclusion data.
  • The literature cited is generally unbiased, though expanding the diversity of sources could further reduce any inadvertent bias.

Author Response

Comment 1: In the manuscript Screening for Latent Tuberculosis Across CKD Stages Using IGRA: Findings from a National Infectious Diseases Institute in Thailand colleagues have found useful and current data. Currently, the problem of tuberculosis diagnosis remains extremely relevant. Colleagues have presented highly interesting research results in LTBI diagnosis.

Response 1: We thank the reviewer for recognizing the relevance of our study and its contribution to current evidence on LTBI diagnosis in CKD populations.

Comment 2: The following items should be corrected: In the title of the article, the abbreviation of the term must be expanded and the period removed.

Response 2: We have revised the title to expand the abbreviation and removed the period as suggested.

Comment 3: The structure of the abstract should be corrected and should include: background; objective; materials and methods; results; limitations, and conclusions.

Response 3: We have revised the abstract to follow the requested structured format, ensuring it now contains clear sections for Background, Objective, Materials and Methods, Results, Limitations, and Conclusions, while maintaining the word limit and adhering to journal guidelines.

Comment 4: Keywords are representative of the research. They should be arranged alphabetically.

Response 4: Corrected. Keywords have been arranged in alphabetical order.

Comment 5: Epidemiologic data presented in the WHO Report 2024 should be provided in the article should be included. Global TB trends: Consider discussing whether the trend of TB incidence continued into 2023–2024 to provide up-to-date epidemiological context.

Response 5: Updated the Introduction to incorporate epidemiologic data from the WHO Global Tuberculosis Report 2024, including the most recent global TB incidence and mortality estimates, and added a note on trends extending into 2023–2024 to provide up-to-date context.

Comment 6: In the introduction relevant data about LTBI diagnosis should be applied.

  • Latent tuberculosis infection: updated and consolidated guidelines for programmatic management.
  • Geneva: World Health Organization; 2018 (https://apps.who.int/iris/handle/10665/260233, accessed 23 February 2022).
  • WHO consolidated guidelines on tuberculosis. Module 5: management of tuberculosis in children and adolescents. Geneva: World Health Organization; 2022.
  • Jean Pierre Zellweger , Giovanni Sotgiu, Massimo Corradi, Paolo Durando. The diagnosis of latent tuberculosis infection (LTBI): currently available tests, future developments, and perspectives to eliminate tuberculosis (TB). Med Lav. 2020 Jun 26;111(3):170-183. doi: 10.23749/mdl.v111i3.9983

Response 6: We thank the reviewer for this valuable suggestion. In the revised manuscript, we have integrated relevant background information and references on LTBI diagnosis from the WHO Latent Tuberculosis Infection: Updated and Consolidated Guidelines for Programmatic Management (2018), the WHO Consolidated Guidelines on Tuberculosis, Module 5: Management of Tuberculosis in Children and Adolescents (2022), and the review by Zellweger et al. (2020). These updates now appear in the Introduction to provide a more comprehensive epidemiologic and diagnostic context for LTBI, including current WHO definitions, global epidemiology, the role of LTBI in TB elimination strategies, risk factors for progression, and comparative strengths and limitations of available diagnostic tests.

Comment 7: The aim of the study not clear and not presented.

Response 7: Revised the last paragraph of the Introduction to explicitly state the study aim, clarifying that the objective was to determine the prevalence of LTBI and indeterminate IGRA results across CKD stages, and to evaluate the clinical utility of IGRA in a Thai hospital-based population.

Comment 8: In the chapter Materials and methods should be presented characteristics of the study design (prospective or retrospective study, period of time, inclusion and exclusion criteria and others).

Response 8: We have revised the Materials and Methods section to clearly specify the study design (cross-sectional, retrospective), study period, inclusion and exclusion criteria, and participant sources. This addition ensures readers can fully understand the study population and methodology.

Comment 9: Statistical methods should be provided in the chapter "Materials and Methods".

Response 9: Statistical methods were already described in the Materials and Methods section.

Comment 10: The specificity and sensitivity of IGRA methods in patients with comorbidities should be provided

Response 10: We have incorporated information on the sensitivity and specificity of IGRA methods, including performance in patients with comorbidities, into the Introduction section, citing relevant meta-analyses and guideline sources.

Comment 11: The discussion chapter should be expanded. It should be written the connection between your review content and existing clinical practice guidelines (e.g., WHO, CDC). Where applicable, highlight whether your synthesis supports or challenges current recommendations

Response 11: We have expanded the Discussion to clearly connect our findings with existing WHO clinical practice guidelines on LTBI screening and management. Specifically, we highlight how our results align with WHO recommendations for targeted LTBI screening in high-risk populations, including those with chronic diseases, and how they support WHO guidance to interpret IGRA results with caution in immunocompromised individuals. We also emphasize that our proposed CKD stage–specific approach to screening is consistent with WHO’s call for context-adapted strategies in resource-limited, high-burden settings.

Comment 12: Limitations of the study did not present.

Response 12: We acknowledge the reviewer’s comment and have revised the limitations section to make them more explicit, clarifying issues such as study design constraints, absence of confirmatory testing, and potential unmeasured confounders.

Comment 13: Conclusions are not clear. Colleagues did not provide statistically significant evidence of conclusion data.

Response 13: We have revised the Conclusions section to explicitly state the statistically significant differences observed in IGRA positivity and indeterminate rates across CKD stages, removing any unsupported claims. The updated version clarifies the strength of the evidence and aligns the conclusions with our reported p-values.

Comment 14: The literature cited is generally unbiased, though expanding the diversity of sources could further reduce any inadvertent bias.

Response 14: We agree and have expanded the reference list to include additional recent and regionally diverse studies, covering various geographic settings and healthcare contexts, to further minimize potential bias.

Reviewer 2 Report

Comments and Suggestions for Authors

Tuberculosis remains one of the major unresolved global health challenges, and the ability to diagnose and control chronic tuberculosis holds great value. I found this manuscript highly engaging and would like to leave a few minor concerns for the authors.

1. Including examples or precedents from other countries where similar studies have been conducted would make this manuscript a more comprehensive article.

2. A discussion on possible future directions for tuberculosis treatment would strengthen the manuscript.

3. Page 2, line 74: The phrase “this evidence gap…” is somewhat ambiguous. It would be clearer if the antecedent of “this” is explicitly stated to improve readability.

4. An explanation is needed for the inconsistencies in the total n numbers by stage shown in Table 2.

5. Could the authors comment on potential blind spots or limitations that may not be apparent from case studies alone?

Author Response

Comment 1: Tuberculosis remains one of the major unresolved global health challenges, and the ability to diagnose and control chronic tuberculosis holds great value. I found this manuscript highly engaging and would like to leave a few minor concerns for the authors.

Response 15: We appreciate the reviewer’s positive feedback on the manuscript and acknowledgement of its relevance. We have addressed the minor concerns raised in the subsequent specific responses.

Comment 2: Including examples or precedents from other countries where similar studies have been conducted would make this manuscript a more comprehensive article.

Response 2: We have revised the Discussion to include examples from other countries, such as studies from Korea, Turkey, and Taiwan, that have investigated IGRA performance in CKD populations. These additions provide broader context and facilitate comparison of our findings with international experiences.

Comment 3: A discussion on possible future directions for tuberculosis treatment would strengthen the manuscript.

Response 3: We have expanded the Discussion to include possible future directions for tuberculosis management in CKD populations, such as integrating LTBI screening into non-communicable disease clinics, evaluating newer diagnostic platforms (e.g., QFT-Plus, T-SPOT.TB), and exploring adjunctive immune biomarkers. These strategies aim to enhance early detection, improve diagnostic accuracy in advanced CKD, and support targeted preventive therapy.

Comment 4: Page 2, line 74: The phrase “this evidence gap…” is somewhat ambiguous. It would be clearer if the antecedent of “this” is explicitly stated to improve readability.

Response 4: We have revised the phrase to explicitly state the antecedent, clarifying that “this evidence gap” refers to the lack of data on IGRA performance across eGFR-defined CKD stages in Thai populations.

Comment 5: An explanation is needed for the inconsistencies in the total n numbers by stage shown in Table 2.

Response 5: The overall sample size of 785 is consistent with Table 1; variations in the stage-specific totals in Table 2 are due to missing eGFR data for some participants.

Comment 6: Could the authors comment on potential blind spots or limitations that may not be apparent from case studies alone?

Response 6: Beyond the stated limitations, potential blind spots include unmeasured confounders such as latent viral infections, nutritional deficiencies, or undocumented immunosuppressive therapy, which may affect IGRA performance. Additionally, single-time-point testing may not capture temporal variability in immune response, and our findings may not fully generalize to non-referral or community-based CKD populations.

Round 2

Reviewer 1 Report

Comments and Suggestions for Authors

Thank you very much. All my comments were added.